



# Evaluation of middle atmosphere temperature and wind measurements and and their disturbance characteristics by meteorological rockets

Yang He[1,2], Jiangping Huang[1], Mingyuan He[2], Zheng Sheng[2]

[1]Beijing Institute of Aeronautics Meteorology, Beijing 100094, China

[2]College of Meteorology and Oceanography, National University of Defense Technology, Changsha 410073, China

*Correspondence to*: Mingyuan He (hmy008@sina.com) and Zheng Sheng (19994035@sina.com)

**Abstract.** It is necessary to carry out the in-situ detection based on the meteorological rocket to deepen the cognitive level of the middle atmosphere environment, though there is still a lack of systematic research on the data accuracy and the physical mechanism affecting the measurement results, which restricts the effective use of rocket data. Based on thermistor and Beidou positioning, combined with temperature correction technology, middle atmosphere temperature and wind measurements from 20-60 km are obtained in northwest China by two meteorological rockets. The detection results are compared with satellite, empirical model and reanalysis data, and the error analysis theory is carried out in combination with the of the drop sounding and atmospheric disturbance characteristics. The results show that the data quality of the rocket detection is ideal, and the variation trend of temperature and wind profile with altitude is consistent with other data. The difference comes from the deviation of the matching data in time and space and the excessive measurement error in the initial fall stage. Also, it is found that the instability of the parachute causes poor positioning data quality and fast falling speed, and eventually cause the measurement error at the corresponding height to be significantly larger. Besides, the profile fluctuation of the first detection is more obvious, which is caused by the fragmentation of the high-altitude gravity wave. Wave dissipation leads to the weakening of atmospheric stability and the generation of denser small-scale layered structures on the profile, making significant wind field changes at the height below through the momentum deposited.

## 1. Introduction

The near space is located in the region of 20-100 km, which can cover the stratosphere, the mesosphere and the low thermosphere. The near-space atmosphere is far from the ground and does not have the weather phenomena common in the troposphere (cyclones, thunderstorms, fronts, etc.), but its unique importance still prompts extensive attention and research. First, the near space is the upper boundary of the troposphere, which can be coupled with the troposphere and affect it from top to bottom. The stratospheric atmosphere, due to its slow evolution characteristics (compared with the troposphere), can provide important information for the prediction of extreme weather and climate in the troposphere (Gray et al., 2018; Jin et al., 2023). For example, the weakening of the stratospheric polar vortex is often a precursor to the occurrence of cold waves in the Northern Hemisphere. Second, the near space is the lower boundary atmosphere of space weather,



which can act as a "display screen" for solar activity, and the influence of solar activity on Earth's weather
and climate can be reflected in it. For example, solar activity can change the ozone in the middle atmosphere
and transmit this change to the troposphere through the action of planetary waves (Krivolutsky et al., 2015).
In addition, the near space is the combination of aerospace and aviation, and changes in the internal
environment will directly affect the flight attitude and effect of aerospace vehicles (Chen et al., 2023; Roney,
40  2007).

Since the near space is showing more and more important value, it is urgent to improve the
understanding of its internal atmospheric environment. The necessary condition to support this demand is to
carry out accurate detection and adequate research. Satellite remote sensing can provide atmospheric profile
data with global coverage, but the detection ability of wind field is still insufficient, and the vertical resolution
of data is rough (Ern et al., 2022; Thies and Bendix, 2011). Lidar and MST (Meso-Stratosphere-Troposphere)
radars can obtain three-dimensional wind fields and temperatures, but the global distribution of detection
sites is limited, and the data quality is affected by atmospheric environment and retrieve accuracy (She et al.,
2003; Daren et al., 2018; Qiao et al., 2020). Flat-floating balloons with zero pressure or overpressure, can
realize continuous detection in the horizontal direction of the stratosphere, but the characteristics of its own
drift in the wind bring the uncertainty of detection, and require strict trajectory control technology (He et al.,
2024; Alexander et al., 2021). Radiosonde balloons can detect meteorological elements with long time series
and high precision, but the highest detection height is generally less than 30 km, and cannot cover higher
airspace (He et al., 2022; Yoo et al., 2020). In contrast, the meteorological rocket sounding is the only in-situ
detection method that can obtain the atmospheric environment in the altitude range of 20~100 km. The
effective evaluation and inspection of rocket detection accuracy is an important prerequisite for the correct
use of this means.
The meteorological rocket sounding mainly includes two methods: falling spheres detection and
thermistor detection. The falling spheres can obtain the atmospheric density profile of 30-100 km, and then
calculate the wind field, temperature and pressure, the thermistor measurement can obtain the atmospheric
temperature from 20 to 60 km, and then calculate the density, pressure, and wind field (Eckermann et al.,
1995; Wang et al., 2006). By comparing with satellite, balloon and reanalysis data, thermistor rockets
launched from Hainan Station and East China Sea have shown good detection results, and the atmospheric
disturbance characteristics in near space are also extracted (Guoying et al., 2011; Song et al., 2024).
Atmospheric density are measured using GPS data on a rigid falling ball and found that the deviation from
the model results was less than 10% (Yuan et al., 2017). Using passive ball falling experiments in northwest
China, in-situ wind field and gravity wave information are analyzed from 30 to 100 km (Ge et al., 2019). A
comprehensive evaluation of the detection accuracy of the TK-1 meteorological rocket is performed and the
reliability is demonstrated (Fan et al., 2013). It can be seen that the current results of near space rocket
detection are still few, encouraging researchers to work in greater depth.
In this paper, two meteorological rockets launched in the northwest of China are used to obtain near
space meteorological detection data from 20 to 60 km, error analysis and accuracy evaluation are carried out,



and wave disturbance characteristics are also extracted. The structure of the paper is as follows: in the second
section, the used data is introduced; in the third section, the temperature correction and error calculation
method are given; in the fourth section, the comparison results of rocket detection profile and reference data
are discussed, in the fifth section, the error analysis is performed; in the sixth section, the characteristics of
wave perturbations and their effects on the background atmosphere are discussed; in the seventh section, the
conclusion and prospect are given.
**2. Rockets instrument and detection principle**
The rockets are from the same batch. The sonde adopts the Beidou navigation and positioning system
with a positioning sampling frequency of 2Hz, the positioning accuracy is 5 m. The temperature sensor is a
beaded thermistor with a size of 0.28 mm×0.4 mm, the measuring range is -90~55 ℃ and the measuring
accuracy is 0.2 ℃. The pressure sensor is a silicon piezoresistive manometer with a sampling frequency of
2Hz, the measuring range is 5~1060 hPa and the measuring accuracy is 0.5 hPa. The main parachute covers
an area of 15 m$^2$ and the sonde has a mass of 0.5kg.
The rocket detection mechanism is shown in Figure 1. The meteorological sonde is carried up by the
rocket, under the action of thrust, it rises at a high speed according to the established trajectory. After the
engine stops working, the rocket uses inertia to continue rising. When the rocket rises near the top of its
trajectory, the parachute carries the sonde and separates from the arrow body. The sonde pulls the parachute
and begins to fall.

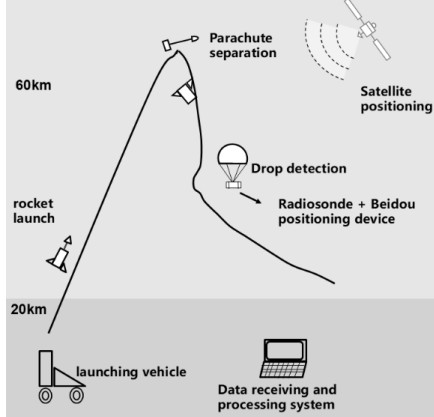


**Figure 1. meteorological rocket detection mechanism.** During this process, the atmospheric parameters are
measured in situ and the data is transmitted down to the ground receiving system. The thermistor sensor is
used to obtain the atmospheric temperature in the altitude range of 20-60 km, and the atmospheric pressure
is obtained layer by layer from iterative calculation based on the base pressure (measured by the pressure
sensor at 20 km). Then the atmospheric density is calculated through the ideal gas equation. The real-time
position coordinates (X, Y, Z) of the sonde are obtained by using the Beidou positioning system, and the first



derivative is obtained by linear fitting after the smoothing position coordinates point by point to calculate the northward, eastward and vertical velocity (represented by $\dot{x}$, $\dot{y}$, and $\dot{z}$). The corresponding acceleration is obtained by quadratic fitting (represented by $\ddot{x}$, $\ddot{y}$, and $\ddot{z}$). Based on the velocity and acceleration information, the meridional, zonal, and synthetic wind are calculated (represented by $W_x$, $W_y$, and $W$), and the wind direction ($\theta$) can be further obtained. The specific calculation formula is shown in (1)-(4)

$$W_x = \dot{x} - \frac{\ddot{x}}{\ddot{z}-g}\dot{z} \tag{1}$$

$$W_y = \dot{y} - \frac{\ddot{y}}{\ddot{z}-g}\dot{z} \tag{2}$$

$$W = \sqrt{W_x^2 + W_y^2} \tag{3}$$

$$\theta = \begin{cases} arctan\left|\frac{W_y}{W_x}\right| + 180°, (W_x > 0, W_y > 0) \\ -arctan\left|\frac{W_y}{W_x}\right| + 180°, (W_x > 0, W_y < 0) \\ -arctan\left|\frac{W_y}{W_x}\right| + 360°, (W_x < 0, W_y > 0) \\ arctan\left|\frac{W_y}{W_x}\right|, (W_x < 0, W_y < 0) \end{cases} \tag{4}$$

The specific calculation process of atmospheric parameters is shown in Figure 2.

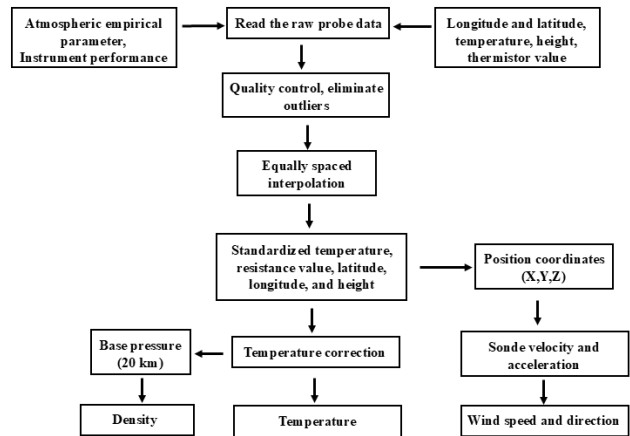

Figure 2. Atmospheric parameters calculation process.

## 3. Data introduction

The data for the two rockets used in this paper are launched in northwest China in the autumn of 2023, the vertical profile of wind velocity (synthetic wind, zonal wind, meridional wind), wind direction, atmospheric temperature, pressure , and density form 20~60 km (effective height interval for analysis) are obtained.



Sounding of the Atmosphere using Broadband Emission Radiometry (SABER) carried on TIMED
satellite, can obtain the vertical profile of atmospheric temperature, pressure, geopotential height, ozone and
other trace gas volume mixture ratio by limb scanning. Here the Level 2A temperature data from Saber
version 2.0 is selected.
MERRA2 (Modern Era Retrospective analysis for Research and Applications version 2) data is an
upgraded version of MERRA data, which is the second generation of high-precision data sets. The data has
a time resolution of 6 h and contains 42 pressure layers ranging from 1000 hPa to 0.1 hPa. The data used in
this paper are zonal wind, meridional wind and atmospheric temperature data. The spatial resolution of the
original data was 0.5°×0.625°.
The NRLMSISE-00 atmospheric empirical model covers the altitude interval from the ground to the
thermosphere (0~1000 km), which can provide reference for the environmental state of the relevant missions
in the space industry. The input parameters include the solar and geomagnetic activity index, date, latitude,
longitude, altitude and local time, and the output elements are the temperature and density profile of the
neutral atmosphere.
When using reanalysis, empirical model, satellite data to compare with rocket detection results, it is
necessary to match the time and location of data effectively. The verification data close to the time (<5 h)
and within a certain deviation range of latitude (<4º) and longitude (<4º) are selected and interpolated to the
same vertical grid points as the data processed by the rocket.
**4. Temperature correction and error calculation**
**4.1 Temperature correction**
During the process of parachute fall, thermistor and the outside atmosphere has been a heat exchange,
in unit time, thermistor internal energy $\Delta E$, self-heating $L$, convection exchange heat $H$, radiation exchange
heat $Q$, viscous exchange heat $M$, lead conduction heat exchange $N$ have the following relationships (Wagner,

1964):

$$\Delta E = L + H + Q + M + N, \quad (5)$$

According to the modified formula given by the World Meteorological Organization on the temperature
detection data of the rocket sonde, formula (5) is expanded to (Organization, 2008):

$$T_\infty = T_f - \frac{rv_f^2}{2c_p} + \frac{m_T C}{Ah}\frac{dT_f}{dt} - \frac{A_m\rho_m\alpha_S J}{Ah} - \frac{\alpha_t\sigma(A_aT_a^4 + A_bT_b^4 + A_cT_c^4)}{Ah} + \frac{\varepsilon\sigma T_f^4}{h} - \frac{Q_c}{Ah} - \frac{W_f}{Ah} \quad (6)$$

Where $T_f$ is the original temperature, $T_\infty$ is the temperature after correction. The heating term $\frac{rv_f^2}{2c_p}$
reflects the influence of heat exchange between the thermistor and its boundary layer on the temperature
indication value. The temperature hysteresis term $\frac{m_T C}{Ah}\frac{dT_f}{dt}$ represents the influence of the hysteresis of
thermistor heat exchange on the temperature indication value. The reflected radiation term $\frac{A_m\rho_m\alpha_S J}{Ah}$
represents the influence of the short-wave solar radiation reflected by the ground and clouds to the sonde on





its temperature indication. The long wave radiation term $\frac{\alpha_t \sigma(A_a T_a^4 + A_b T_b^4 + A_c T_c^4)}{Ah}$ represents the influence of
radio frequency radiation and infrared radiation in the environment of the sonde on the temperature indication.
The external radiation term $\frac{\varepsilon \sigma T_f^4}{h}$ represents the influence of the thermal radiation of the sensor to the sonde
on its temperature indication. The structural heat conduction term $\frac{Q_c}{Ah}$ represents the influence on the
thermistor indication due to the thermal conduction of the sonde support to the thermistor. Measuring current
heating term $\frac{W_f}{Ah}$ indicates the amount by which the temperature indication of the resistance changes due to
the heating of the current. The sonde takes shading measures to ignore the direct solar radiation. The
meanings of each item in equation (6) are shown in Table A1.
**4.2 Error calculation**

Temperature measurement error is composed of thermistor static calibration error $\sigma T_1$, temperature

error caused by position error $\sigma T_2$, and temperature correction error $\Delta T_3$ (Wagner, 1964, 1961), the
calculation formula is as follows:
$$\delta T = \sqrt{\sigma T_1^2 + \sigma T_2^2 + \Delta T_3^2}, \tag{7}$$

$\sigma T_1$ and $\sigma T_2$ are the systematic errors of the instrument, which are fixed values in calculation, $\Delta T_3$ is
the residual error after temperature correction (Eq. 6), and the formula is calculated as:
$$\Delta T_3 = \Delta(T_\infty - T_f) = \Delta\left(-\frac{rv_r^2}{2c_p}\right) + \Delta\left(\frac{m_T C}{Ah}\frac{dT_f}{dt}\right) + \Delta\left(-\frac{A_s \alpha_s J}{Ah}\right) + \Delta\left(-\frac{A_m \rho_m \alpha_s J}{Ah}\right) +$$

$$\Delta\left(-\frac{\alpha_t \sigma(A_a T_a^4 + A_b T_b^4 + A_c T_c^4)}{Ah}\right) + \Delta\left(\frac{\varepsilon \sigma T_f^4}{h}\right) + \Delta\left(-\frac{Q_c}{Ah}\right) + \Delta\left(-\frac{W_f}{Ah}\right), \tag{8}$$

Wind speed error is composed of systematic error and random error. Systematic error is written as:
$$\begin{cases} \Delta W_x = \Delta\dot{x} - \frac{\dot{z}}{\ddot{z}-g}\Delta\ddot{x} - \frac{\ddot{x}}{\ddot{z}-g}\Delta\dot{z} + \frac{\ddot{x}\dot{z}}{(\ddot{z}-g)^2}\Delta\ddot{z} \\ \Delta W_y = \Delta\dot{y} - \frac{\dot{z}}{\ddot{z}-g}\Delta\ddot{y} - \frac{\ddot{y}}{\ddot{z}-g}\Delta\dot{z} + \frac{\ddot{y}\dot{z}}{(\ddot{z}-g)^2}\Delta\ddot{z} \end{cases}, \tag{9}$$

Random error is written as:
$$\begin{cases} \sigma_{W_x}^2 = \sigma_{\dot{x}}^2 + \left(\frac{\dot{z}}{\ddot{z}-g}\sigma_{\ddot{x}}\right)^2 + \left(\frac{\ddot{x}}{\ddot{z}-g}\sigma_{\dot{z}}\right)^2 + \left[\frac{\ddot{x}\dot{z}}{(\ddot{z}-g)^2}\sigma_{\ddot{z}}\right]^2 \\ \sigma_{W_y}^2 = \sigma_{\dot{y}}^2 + \left(\frac{\dot{z}}{\ddot{z}-g}\sigma_{\ddot{y}}\right)^2 + \left(\frac{\ddot{y}}{\ddot{z}-g}\sigma_{\dot{z}}\right)^2 + \left[\frac{\ddot{y}\dot{z}}{(\ddot{z}-g)^2}\sigma_{\ddot{z}}\right]^2 \end{cases}, \tag{10}$$

$g$ is the gravity acceleration, $\Delta\dot{x}$, $\Delta\dot{y}$ and $\Delta\dot{z}$ are velocity fitting deviations, $\Delta\ddot{x}$, $\Delta\ddot{y}$, and $\Delta\ddot{z}$ are
acceleration fitting deviations, $\sigma_{\dot{x}}$, $\sigma_{\dot{y}}$, and $\sigma_{\dot{z}}$ are speed random errors, $\sigma_{\ddot{x}}$, $\sigma_{\ddot{y}}$, and $\sigma_{\ddot{z}}$ are acceleration
random errors.
The total error of wind speed and direction is calculated as follows:
$$\begin{cases} \delta W_\varepsilon = \sqrt{\delta W_{x\cdot\varepsilon}^2 + \delta W_{y\cdot\varepsilon}^2} \\ \delta G = \frac{180}{\pi}\sqrt{\left(\frac{W_x \cdot \delta W_{y\cdot\varepsilon}}{W_x^2 + W_y^2}\right)^2 + \left(\frac{W_y \cdot \delta W_{x\cdot\varepsilon}}{W_x^2 + W_y^2}\right)^2}, \end{cases} \tag{11}$$





173 Where $\delta W_{x\cdot\varepsilon} = \sqrt{\sigma_{W_x}^2 + \Delta W_x^2}$ is the meridional wind synthesis error, and $\delta W_{y\cdot\varepsilon} = \sqrt{\sigma_{W_y}^2 + \Delta W_y^2}$ is the

174 zonal wind synthesis error.

## 175 5. Comparison of rocket detection results with reference data

### 176 5.1 Data quality and trajectory analysis

177 The two rockets are referred to as HJ-1 and HJ-2, respectively. HJ-1 is launched at 9:00 UTC on the

178 first day, and HJ-2 is launched at 5:00 UTC on the next day.

179 The time-height curves of HJ-1 and HJ-2 are shown in Figure 3 (top). The actual detection altitude of

180 HJ-1 is about 74 km, the ascent time is about 2 minutes, and the fall time (from the highest point to an altitude

181 of 20 km) is 25 minutes. HJ-2 can reach a maximum altitude of 76 km, the ascent time is about 2 minutes,

182 and the fall time is 31 minutes. Taking the launch point as the central point, the horizontal motion trajectory

183 of the ascending stage of rocket launch and the sonde/parachute drift stage are plotted as shown in Figure 3

184 (below). When the rocket is launched, it rises basically eastward, and after reaching the highest point, the

185 sonde drifts eastward as it falls with the parachute, which is determined by the background wind field over

186 the area (the trajectory indicates that the entire layer is dominated by westerly winds). The sonde remained

187 within 100 km from the launch point during the entire detection process (from the beginning of the launch to

188 the 20 km falling height).

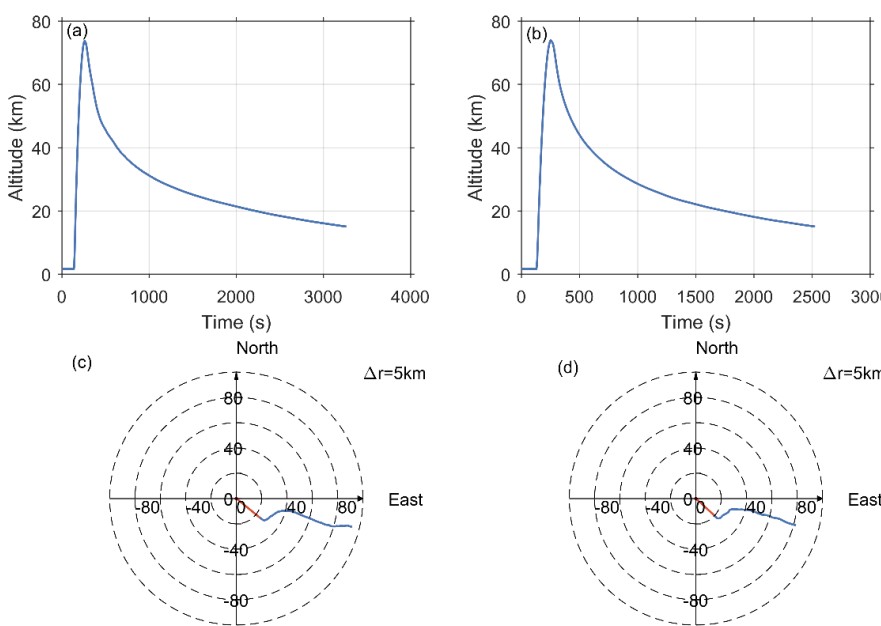


**Figure 3. Time-altitude curves of (a) HJ-1 and (b) HJ-2, and horizontal motion trajectory of (a) HJ-1 and (b) HJ-**
**2 (red for rocket ascent, blue for sonde/parachute drift).**





In order to further analyze the trajectory characteristics of the sonde during its fall, the vertical
distribution of zonal velocity (vx), meridional velocity (vy) and vertical velocity (vz) are shown in Figure 4.
The zonal velocity of the two rockets is positive, and the meridional velocity gradually changes from positive
to negative, which corresponds to the characteristics of the falling trajectory drifting first to the northeast and
then to the southeast in Figure 3. It is worth noting that there is an obvious disturbance characteristic (denser
small scale layered structure) of vertical velocity for HJ-1, compared with that of HJ-2. After the same data
processing method, the obvious difference of vz profile roughness may reflect the great difference of
disturbance in the vertical direction at high altitudes.

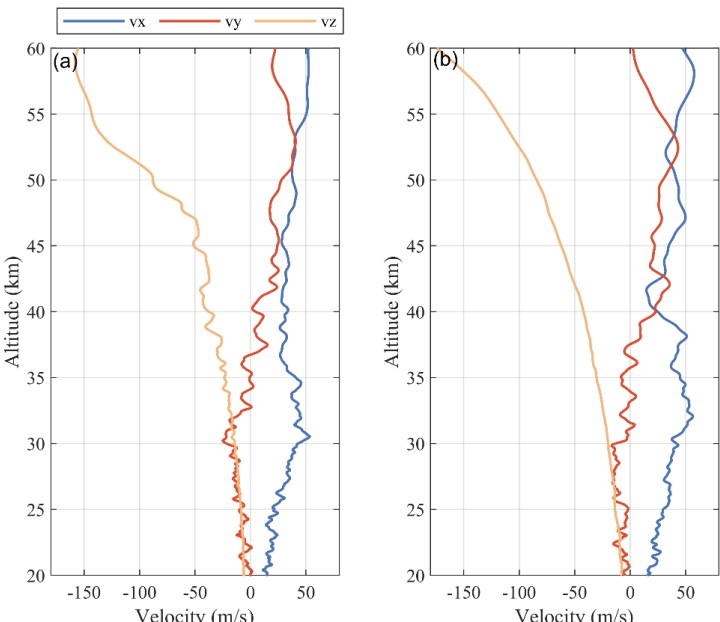

**Figure 4. Velocity-altitude curves of (a) HJ-1 and (b) HJ-2.**
**5.2 Wind and temperature measurements**
Figure 5 shows the comparison of the zonal and meridional winds obtained by the two rockets with the
MERRA2 data. Before the launch of the rocket, the balloon sounding is also carried out. In the detection
altitude range of the balloon, the wind speed and direction from the rocket are in good agreement with the
balloon profile, and the details of the disturbance are basically consistent (Figure A1), indicating the
reliability of the retrieved wind field results. The meridional winds of the two rockets both reach the
maximum value near 50 km, exceeding 40 m/s. As reflected in Figure 3, in the initial stage of fall after the
rocket body-parachute separation, the trajectory turns from south to north, which proves that the strong
meridional winds dominate at high altitudes.




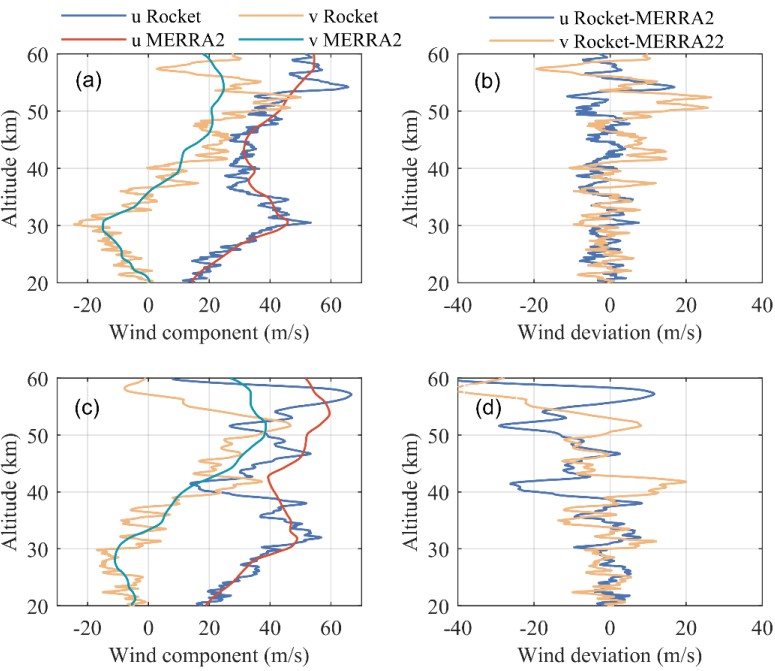

**Figure 5. (a) The vertical distribution of zonal wind and meridional wind of HJ-1, (b) the difference of HJ-1**
**velocity component with MERRA2, (c) the vertical distribution of zonal wind and meridional wind of HJ-2, and**
**(d) the difference of HJ-2 velocity component with MERRA2.**
HJ-1 and MERRA2 have basically the same variation trend of wind speed components at the altitude
of 20~60 km, and the zonal wind deviation is relatively small in the whole altitude, while the meridional
wind deviation has large positive and negative fluctuations between 50~60 km. The average deviation
(absolute value) of the zonal wind at the whole altitude is 3.3 m/s, and that of the meridional wind is 5.4 m/s.
In contrast, the measured wind of HJ-2 has greater fluctuation than that of MERRA2, and the deviation
between the two increases significantly above 40 km. The average deviation of zonal wind is 7.5 m/s and
that of meridional wind is 7.6 m/s. In the altitude range of 20~45 km, the variation trend of wind speed is
consistent. At higher altitudes, the measured wind speed of the rocket can show more significant fluctuation
characteristics. There are maximum wind speed areas near 30 km and 55 km for both the two rockets, and
the maximum near 55 km is difficult to reflect in the MERRA2 data. This indicates that the reanalysis data
may lack observation results for assimilation at higher altitudes, and the difference of wind field in the upper
stratosphere is obviously greater than that in the lower stratosphere even in the close spatiotemporal range.
Considering that the output from the model tends to reflect the average trend, and the transient results of a
single detection are more prominent, it is reasonable to have differences between the rocket detection and the
model.
Figure 6 shows the vertical distribution of temperature from rocket, SABER, MSISE, and MERRA2
data and the corresponding deviation from them. Results before and after temperature correction and
corresponding sub-term correction amount are shown in Figure A2, the temperature correction is larger above





50 km, and gradually decreases below 50 km. The corrected temperature is smaller than original temperature
in the entire height. According to the maximum temperature, the stratopause height measured by the rocket
(the height of the inflection point) is around 47 km. The stratopause height is consistent with other reference
data for HJ-1, but shows some differences for HJ-2. The temperature profiles of the four data have a consistent
trend from 20 km to 50 km, with small deviation. The deviation between the reference data and the rocket
detection results increases above 50 km. In this interval, the temperature deviation between HJ-1 and MISIS
is the smallest, while the difference between HJ-2 and SABER is the smallest. It is worth noting that the
temperature deviation of HJ-1 above 57 km has a sharp trend, which may be resulted from its measurement
error (discussed later). The difference of data comparison may be due to the following reasons: 1) There are
deviations in the position and time of the reference data matching with the rocket; 2) The results of the model
reflect the average over time and space, which is indeed different from the single-point profile.

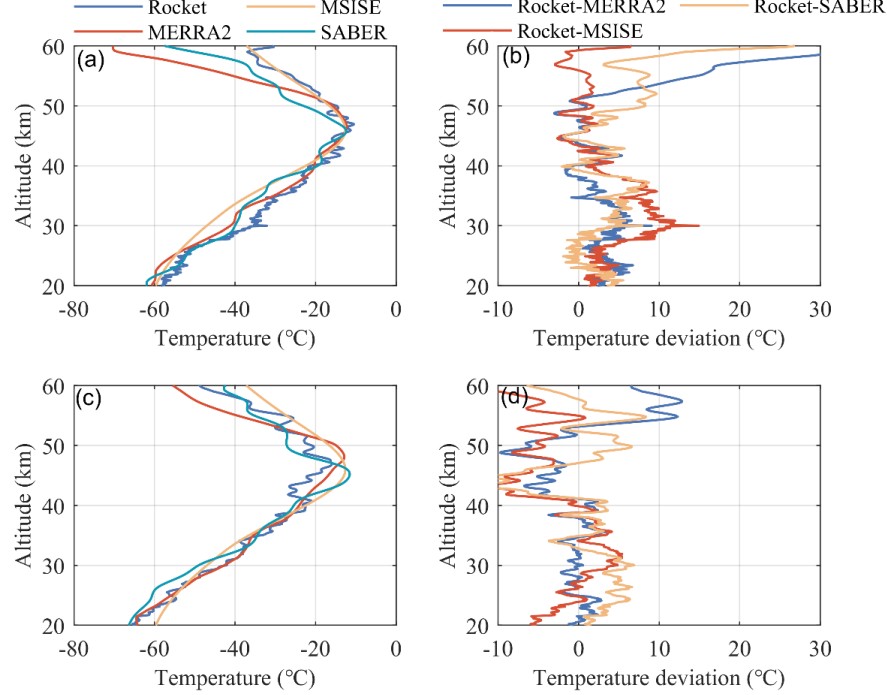

**Figure 6. (a) The vertical distribution of temperature for HJ-1, (b) the difference of HJ-1 temperature with**
**MERRA2, MSISE, and SABER, (c) the vertical distribution of temperature for HJ-2, and (d) the difference of**
**HJ-2 temperature with MERRA2, MSISE, and SABER.**
**6. Error analysis**
Temperature and wind measurement errors of HJ-1 and HJ-2 can be obtained according to Eq. 8 and Eq.
11, as shown in Figure 7. Systematic and random errors of wind speeds are shown in Figure A3 and Figure
A4, respectively. The atmospheric temperature error level (regional average) of HJ-1 is 0.31 ℃, 0.53 ℃ and
5.5 ℃ at 20-30 km, 30-50 km and above 50 km, while that of HJ-2 is 0.24 ℃, 0.55 ℃ and 1.75 ℃. The wind





speed error level of HJ-1 is 0.63 m/s, 1.12 m/s and 4.95 m/s at 20-30 km, 30-50 km and above 50 km, while
that of HJ-2 is 0.38 m/s, 1.19 m/s and 4.0 m/s. The wind direction error levels of HJ-1 are 0.81°, 1.08° and
3.15° at 20-30 km, 30-50 km and above 50 km, respectively, while that of HJ-2 is 0.54°, 1.11° and 4.25°.
According to Eq. 9 and Eq. 10, when the vertical acceleration and vertical velocity are too large, the
denominator $\ddot{z} - g$ decreases and the numerator $\ddot{z}$ increases, which can obviously affect the results of
systematic error and random error. In the whole detection section, the same smooth fitting points are used,
so the velocity error is consistent. However, due to the large jump of the positioning data, the acceleration
ratio in the inertial velocity will also jump. When the falling velocity is large, the product will also increase,
resulting in a significantly larger error margin at the high altitudes.

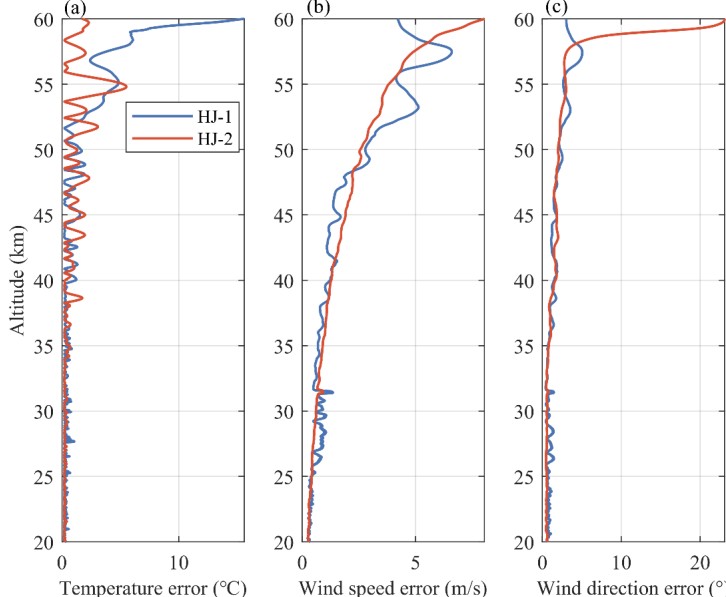


**Figure 7. Error-height curves of (a) temperature, (b) wind speed, and (c) wind direction for HJ-1 and HJ-2.**
The original temperature vertical gradient and vertical acceleration of HJ-1 and HJ-2 are shown in
Figure 8. In the initial falling stage (50-60 km) after the parachute separation, the falling speed is too large,
and the acceleration fluctuates significantly in this height range. The vertical acceleration of HJ-1 has two
peaks between 50-60 km, and there are also maximum values in the corresponding height of the wind speed
random error and systematic error profile (Fig. A3). The vertical acceleration of HJ-2 increases rapidly above
50 km, which also corresponds to the increasing trend of wind speed component in systematic error and
random error. According to the error equation, the measurement error of wind speed depends largely on the
velocity error and acceleration error. At the same time, the temperature error is also related to the vertical
gradient of the measured temperature indication value (in HJ-1, the obvious gradient deviation above 58 km
and its ratio to the convective heat exchange coefficient cause the temperature error to increase sharply),





which is also the reason why the temperature error and wind field error in Figure 7 have inconsistent trends.
Through the above analysis, we believe that the intensity of vertical acceleration fluctuation directly affects
the error results of wind field measurement. At high altitudes (near to 60 km), the parachute swing is large,
and the data reception is not stable, resulting in the relatively low positioning data quality and the large
position error, which finally lead to the relatively large wind field error. As the detection height gradually
decreases, the positioning data quality increases and the measurement error decreases gradually as the
parachute falls steadily.

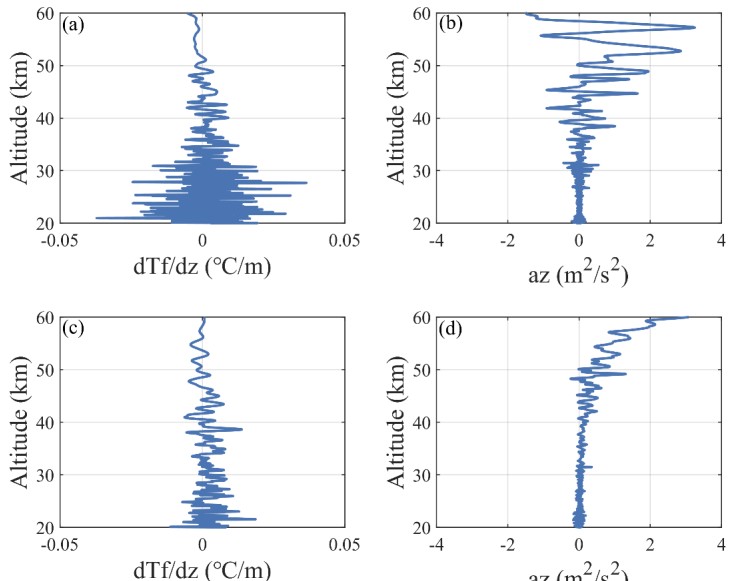

**Figure 8. (a) Temperature gradient-height curve and (b) vertical velocity-altitude curve for HJ-1, (c) Temperature**
**gradient-height curve and (d) vertical velocity-altitude curve for HJ-2.**
**7. Disturbance characteristic analysis**
**7.1 Wave energy and background field analysis**
Due to the lack of measured wind field data at high altitudes (30-60 km), the fine structure cognition of
wind disturbance at corresponding interval is not sufficient. Many of the sharp peaks in the wind profile
captured by balloon and rocket detections are real perturbations in the atmosphere (Figure 5 and Figure A1),
which are smoothed out in the reanalysis. In other words, using rocket data may be more suitable for
analyzing wave disturbance characteristics at high altitude, since reanalysis data failed to capture these details.
The apparent differences in vertical velocity and acceleration of the sonde during its fall (Figure 8) also
indicate significant differences in upper atmospheric disturbances. By analyzing the atmospheric background
state and gravity wave (GW) information, we compare the difference characteristics of atmospheric
disturbance in two detection processes.



GWs are generated by the excitation source at the lower atmosphere, and their amplitudes increase

gradually as the atmospheric density decreases during upward propagation. Wind shear is an important
disturbance source of high-altitude GWs, which can cause GWs to be generated or broken (Larsen, 2002;
Larsen and Fesen, 2009). Vertical wind shear can be calculated by the following formula:

$$\frac{dU}{dz} = \sqrt{\left(\frac{du}{dz}\right)^2 + \left(\frac{dv}{dz}\right)^2},\tag{12}$$

Buoyancy frequency $N$ can reflect the unstable state of the atmosphere. $N^2 > 0$ is the static stable state,

and $N^2 < 0$ is the static unstable state. The square buoyancy frequency can be calculated by the following
formula:

$$N^2 = \frac{g}{T}\left[\left(\frac{dT}{dZ}\right) + \frac{g}{c_p}\right],\tag{13}$$

The gradient Richardson number $R_i$ can reflect the ratio of buoyancy work term to shear stress work

term, which can be obtained by the ratio of the square of buoyancy frequency to the square of wind shear:

$$R_i = \frac{N^2}{\left(\frac{du}{dz}\right)^2 + \left(\frac{dv}{dz}\right)^2},\tag{14}$$

Atmospheric GWs can be regarded as superimposed disturbances to the background field. First, a 20-

point sliding average is performed on the profile interpolated with equal spacing (50 m interval) to eliminate
errors caused by random motion and turbulence. Then the smoothing profile is fitted by fifth-order
polynomial to get the background profile. After the background profile is removed, high-pass filtering with
a cut-off wavelength of 10 km is performed to obtain the disturbance profile caused by GWs. The kinetic
energy $E_k$ of GW is calculated by the following formula:

$$E_k = \frac{1}{2}\left(u'^2 + v'^2\right),\tag{15}$$

Where $u'$ and $v'$ are the disturbance components of the zonal and meridional wind field caused by GWs,

respectively.

In the error analysis, considering that the error becomes significant above 55 km (Figure 7), the height

interval selected for disturbance analysis here is 20~55 km. The vertical distribution of wind shear, square
buoyancy frequency, Richardson number and kinetic energy obtained according to HJ-1 and HJ-2 detection
results are shown in Figure 9. The wind shear of HJ-1 has the first peak (strongest) near 45-55 km and the
second peak near 30-40 km, while the wind shear peak of HJ-2 is between 30-40 km. The buoyancy frequency
is positive at the whole altitude, indicating that the atmosphere is statically stable, but there is a tendency to
decrease with the increase of altitude. HJ-1 has a buoyancy frequency minimum (even close to 0) between
45 and 55 km, corresponding to large wind shear, resulting in a relatively concentrated area of $R_i < 0.25$,
indicating strong dynamic instability. In contrast, HJ-2 has a smoother profile with smaller wind shear and
larger buoyancy frequency, resulting in fewer dynamic instability regions. For HJ-1, the peak kinetic energy
of GW is above 50 km, corresponding to the maximum value region of wind shear, and the dynamic
instability region is relatively concentrated, indicating that Kelvin-Holtzmann instability has produced strong
high-altitude wave disturbance. Below 50 km, the GW energy of HJ-1 is significantly smaller than that of
HJ-2, which is mainly due to the attenuation of zonal wave disturbance (Figure A5).





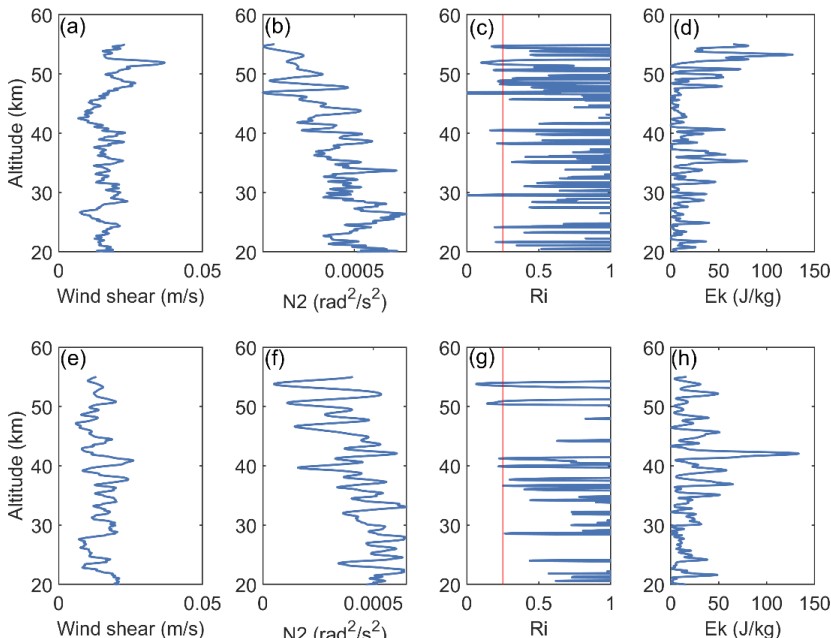

**Figure 9.** (a) wind shear, (b) square buoyancy frequency, (c) Richardson number and (d) kinetic energy of HJ-1,
and (e) wind shear, (f) square buoyancy frequency, (g) Richardson number and (h) kinetic energy of HJ-2.
**7.2 Spectral analysis**
Lomb-Scargle spectrum analysis is performed to the disturbance profile of the synthesized wind speed,
and the vertical wave-number spectrum caused by GWs are obtained, as shown in Figure 10. The amplitudes
of GWs measured by HJ-1 are significantly weaker than those measured by HJ-2, and the vertical
wavelengths of the dominant GWs (amplitudes greater than 90% confidence) are more dispersed, with scales
ranged from 1.7 km to 7.1 km present. In contrast, the dominant GWs measured by HJ-2 have stronger
amplitudes and are concentrated at wavelengths around 4-6 km and 2.9 km.





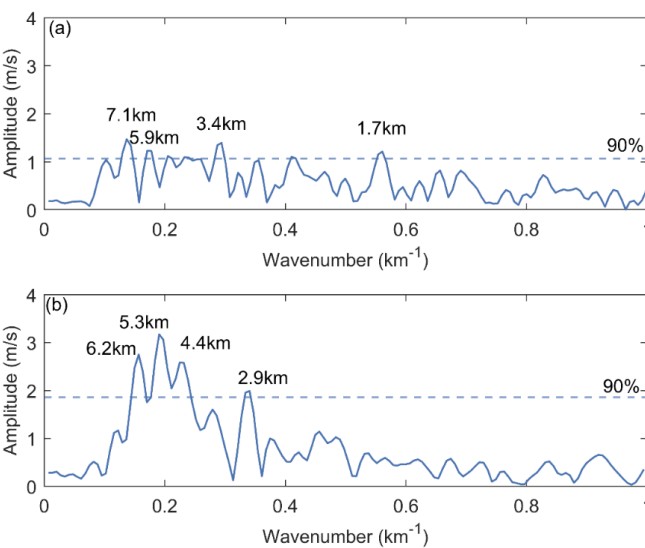

**Figure 10. Gravity wave information for (a) HJ-1 and (b) HJ-2 obtained from the disturbance profile of the wind field, dashed lines represent 90% confidence, and dominant wavelengths with amplitudes above this threshold are labeled.**

Through atmospheric instability analysis and GW spectrum analysis, we found that the atmospheric disturbance for HJ-1 is more complex. The GW breaks, resulting in enhanced turbulent activity (more dynamic unstable regions), which also leads to a significant reduction in stratification stability (reduced buoyancy frequency) with more small-scale stratification (Held et al., 2019; van Haren et al., 2015). The GW kinetic energy can be reduced and the amplitude corresponding to the dominant wavelength decreases. Therefore, compared with HJ-2, the measured temperature and wind field profile of HJ-1 have more obvious fluctuations and a denser small-scale layered structure. In addition, the wave energy of HJ-1 is significantly lower than that of HJ-2 in the range of 40-50 km (Figure 9), which is considered to be the main region where wave dissipation occurs. At this time, the zonal and meridional winds of HJ-1 are also smaller than those of HJ-2 between 40-50 km, while their trends and magnitudes below 40 km are indeed similar (Figure 5), which further indicates that wave dissipation weakens the local winds. The fragmentation and dissipation of GWs in the upper stratosphere can reasonably explain the difference of detection profiles in adjacent two days.

Aiming at the disturbance of the resultant wind speed, the vertical wavenumber spectrum analysis is carried out at the height intervals of 20-30 km, 30-40 km, 40-50 km and 20-50 km respectively, and the results are shown in Figure 11. Pre-whitening and Hanning window smoothing are also be performed on the disturbance profile. For specific technical details, refer to previous research (Dewan and Grossbard, 2000; He et al., 2020a). The fitting intervals are selected between the scales of 3.5 km and 200 m, corresponding to the reasonably resolvable small-scale fluctuations and the noise level eliminated by smoothing filtering, respectively. For HJ-1, due to the wave breaking during the upward propagation of the GW, the spectrum slope is lower (-1.81) at the height of 20-50 km, which is much lower than the theoretical spectrum of -3



(Eckermann, 1995) and close to the turbulence spectrum slope of -5/3 (Kolmogorov, 1941). The spectrum
slope shows the smallest value (-1.40) at the height of 40-50 km, where the GW breaking occurs. The above
spectral structure characteristics indicate the generation of turbulence after gravity wave breaking, and the
energy transfer from large scale region to small scale region (Lu and Koch, 2008; He et al., 2020b). In contrast,
GW energy from HJ-2 is stronger, but it does not become unstable, so the spectral slope is closer to the
universal spectrum of -3.

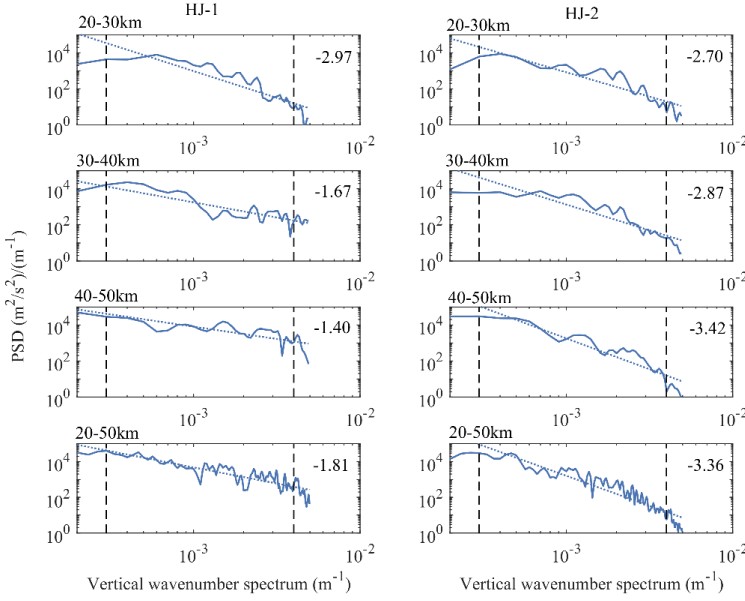


**Figure 11. The vertical wavenumber spectra of (left) HJ-1 and (right) HJ-2 at intervals of 20-30 km (top),**
**30-40 km (middle 1), 40-50 km (middle 2) and 20-50 km (bottom), respectively. The dashed blue line is the fitting**
**slope, and the dashed black line is the fitting interval.**
**7.3 Wave dissipation revealed from Stokes parameter method and ERA5 results**
In order to further prove that the GW at the height of 40-50 km in the detection of HJ-1 has broken up,
Stokes parameter method (Vincent et al., 1987; Eckermann., 1996) is used here to extract the typical
characteristic parameters of the GW. The main realization path is as follows: Fourier transform is applied to
the zonal wind and meridional wind disturbances, and corresponding real and imaginary parts are obtained
respectively. Then four Stokes parameters I, D, P and Q are calculated, and information such as scale,
propagation and frequency of polychromatic gravity waves can be further obtained. The specific method can
be referred to the previous paper (He et al., 2022).
Considering that the wave breaking mainly occurs below 50 km, the GW parameters are calculated for
the two height intervals of 40-50 km and 20-50 km, corresponding to disturbance information in the local
and entire height range, respectively. The kinetic energy, horizontal wavelength, intrinsic frequency, vertical
group velocity and horizontal propagation direction extracted from the two detections are shown in Table 1.





For a local wave disturbance (40-50 km), there is a low-frequency GW of HJ-1, with an intrinsic frequency
(the ratio of wave frequency to inertial frequency) of 2.53. The order of wavelength, kinetic energy and
vertical group velocity is within a reasonable range. In contrast, the intrinsic frequency and vertical group
velocity of HJ-1 are abnormally large, while the horizontal wavelength are abnormally small, which should
belong to the omitted cases. The outliers of the characteristic parameter also reflect the breaking of GWs in
this region from the perspective of abnormal high frequency waves (Fritts and Alexander, 2003), meaning
that GWs can no longer maintain their normal state and dissipate. For the entire wave disturbance (20-50
km), HJ-2 has no obvious wave breaking, and the parameters such as wavelength and frequency are close to
the local disturbance, which means a consistent wave propagation process throughout the entire height. In
contrast, the wavelength and kinetic energy of the entire wave disturbance of HJ-1 are smaller than that of
HJ-2 due to local wave breaking. The wave propagation direction of HJ-2 is significantly different in the
entire and local ranges, possibly due to significant wind speed changes near 40 km (Figure 5c).
Table 1. Gravity wave parameters extracted by Stokes parameter method

| detection zone | kinetic energy (J/kg) | horizontal wavelength (km) | intrinsic frequency (w/f) | vertical group velocity (m/s) | horizontal propagation direction |
|---|---|---|---|---|---|
| HJ-1 (40-50 km) | 13.55 | 1.08 | 434.71 | 15.77 | 138° |
| HJ-2 (40-50 km) | 28.16 | 235 | 2.53 | 0.088 | 215° |
| HJ-1 (20-50 km) | 14.38 | 140 | 3.74 | 0.12 | 145° |
| HJ-2 (20-50 km) | 20.79 | 296 | 2.53 | 0.10 | -36° |

Although the time interval of the two detections is one day apart, considering that the momentum
deposition of GW to the mean flow is continuous and slow (Liu et al., 1999), the comparison of the wind
field results from the two detections may still indicate the effect of GW drag. For the local wave breaking of
HJ-1 (40-50 km), the propagation direction is northwest (the degrees in Table 1 represent angle measured
anticlockwise from x axis), and the deposited momentum produces negative drag (deceleration) on the zonal
wind and positive drag (acceleration) on the meridional wind. Compared with the earlier detection,
significantly stronger meridional wind and significantly weaker zonal wind can be seen near 40 km in the
later detection (Figure 5). This suggests that the drag effect of local wave breaking through deposited
momentum is captured at an altitude of 40 km by HJ-2, and the acceleration of tens of meters per day is also
reasonable (Li et al., 2022).
In order to further support the wave breaking at high altitude during HJ-1 detection, ERA5 data is used
to plot the longitude-latitude cross section of vertical velocity at 3hPa (near 41km) in the corresponding
region (10° longitude × 5° latitude), as shown in Figure 12. HJ-1 detection is close to 09UTC (launched at
9.5UTC) on the first day (T), and HJ-2 detection is close to 06UTC (launched at 5.5UTC) on the second day
(T+1). In this pressure layer, the vertical velocity (w) has an obvious alternating positive and negative
perturbation, which indicates the GW activity. For the first day of detection, at 03UTC and 06UTC, the
northwestward movement of the GW is observed. At 09UTC, there is a distinct wave breaking (purple
rectangular box). For the second day of detection, at 03UTC and 06UTC, the southwestward movement of





the perturbation peaks can be observed, and no wave dissipation occurs in the corresponding region. Both
the time of wave dissipation and the direction of wave propagation are consistent with the results calculated
by Stokes parameter method from rocket data (Table 1), which further proves the reliability of the results.

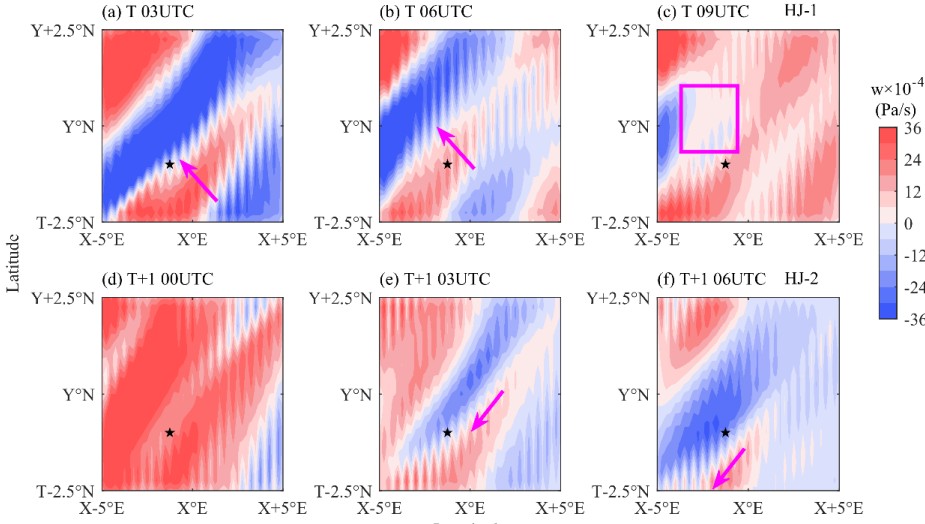


**Figure 12.** **Regional distribution of ERA5 vertical velocity (w) at 3hPa for (a) 03UTC, (b) 06UTC, and (c) 09UTC**
**on the first day, and (d) 00UTC, (e) 03UTC, and (f) 06UTC on the second day, with the five-pointed star**
**representing the rocket detection position. The purple arrow represents the direction in which the wave travels,**
**and the purple rectangular box represents the region where the wave dissipates occurs. The launch point of rocket**
**is (X°E,Y°N).**

## 8. Summary


In this study, the detection effects and data quality of two meteorological rocket launched in the
northwest of China in the autumn of 2023 are analyzed. First, using the modified temperature correction
model and wind field retrieval algorithm, the atmospheric temperature, pressure, density, wind speed and
wind direction measured by the rocket are obtained, and compared with the matched reanalysis, satellite and
empirical model data. Then, using the methods of error transfer and error synthesis, the measuring error and
data accuracy from rocket detection are calculated, and the influence effect is evaluated. Finally, the
characteristics of atmospheric instability and GW activity are analyzed and discussed. The main conclusions
are as follows:
(1) The data acquisition rate of the two rockets is ideal, and the motion trajectory of the ascending and
falling stages is normal and smooth. It is a relatively successful detection experiment, which can obtain good
quality meteorological data in the range of 20 to 60 km.
(2) The rocket detection wind field and MERRA2 wind field have a good agreement below 40 km, and
the deviation becomes larger above 40 km. The temperature detection data below 50 km has a good agreement
with MERRA2, MISIS and SABER, and the deviation above 50 km begins to increase. The difference in



time and space of the matching data, as well as the difference between the model average and the
instantaneous detection may be the source of the result bias.
(3) Below 50 km, the wind measurement error and temperature measurement error are maintained at a
small level, less than 2 m/s and 1.8 ℃, respectively. Above 50 km, the error begins to increase. This is
because in the early stage of fall, the parachute swing is large, so the position error is large. The data reception
is not stable, and the relative speed of the sensor and the air is too large. The above phenomenon leads to the
large error.
(4) The difference in the intensity of GWs causes the obvious difference in vertical velocity of the
dropsonde. For HJ-1, the amplitude of GWs over this region is reduced, and turbulent activity is enhanced,
resulting in reduced stability of atmospheric stratification and a denser small scale hierarchical structure on
the profile. For HJ-2, the stratification stability of the upper atmosphere is stronger. GWs are more stable and
less likely to break, allowing the amplitude to grow to a larger degree.
(5) The local breaking of GWs at 40-50 km can be captured ideally from HJ-1. The GWs deposited
momentum and energy to the mean flow, and the effect of the wave drag changed the wind field structure
below, making HJ-2 with one day delay can detect significant wind field changes near the altitude of 40 km.
This reflects the forcing effect of wave dissipation on the background wind field through the observation
results. The results of ERA5 data further support the wave dissipation and propagation characteristics
extracted by rocket data.
The analysis shows that due to the high vertical resolution and in-situ detection method, the rocket drop
sounding can capture the fine structure of the atmosphere close to the real state. Accurate and detailed wind
field results are very valuable, especially in the region above 30 km. The large measurement error above 55
km also indicates that it is necessary to improve the data reception quality at the beginning of the drop, and
optimize the high-altitude parachute opening and stability control technology to improve the detection
accuracy. The existence of atmospheric gravity wave causes the local feature difference of the detection
profile, meaning that the high-altitude disturbance characteristics need to be considered in the rocket
detection. This study can support the application of the wave dissipation theory in the upper stratosphere by
using the rocket data, while the corresponding ideal observation examples at this altitude are scarce. More
subsequent rocket detections are also encouraged to be carried out, thereby improving the cognition level of
the near-space atmospheric environment in multi-regions and multi-time.





**Appendix A**

Table A1. variable meaning in the equation (6)



| variable | meaning | variable | meaning |
|---|---|---|---|
| $r$ | Thermistor boundary layer temperature recovery coefficient | $\alpha$ | Stefan-Boltzmann constant |
| $v_r$ | The speed at which the thermistor moves with respect to the atmosphere | $J$ | Solar constant |
| $c_p$ | Specific heat capacity of air at constant pressure | $\alpha_t$ | Absorption rate of long wave by thermistor |
| $m_T C$ | Heat capacity of a thermistor | $A_a/A_b/A_c$ | The effective area of the thermistor receiving upper/body/lower bound atmospheric long wave radiation |
| $A$ | The surface area of the thermistor | $T_a/T_b/T_c$ | Equivalent blackbody temperature of upper/body/lower bound atmospheric long-wave radiation source |
| $h$ | Convective heat exchange coefficient between thermistor and air | $\varepsilon$ | Thermistor the emissivity of thermal radiation |
| $\dfrac{dT_f}{dt}$ | The rate of change of temperature with time | $Q_c$ | Heat conduction coefficient |
| $A_m$ | The area of the thermistor reflected by the ground and clouds | $W_f$ | Current work coefficient |
| $\rho_m$ | Combined reflection coefficient of ground and cloud | | |


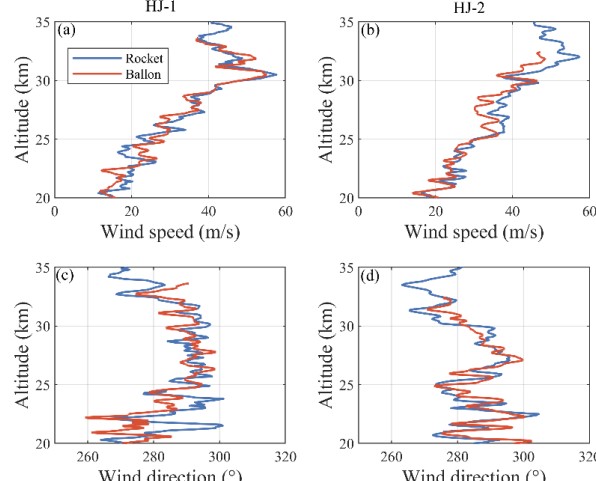


**Figure A1. Comparison of wind speeds measured by rockets and balloons for (a) HJ-1 and (b) HJ-2,and comparison of wind directions measured by rockets and balloons for (c) HJ-1 and (d) HJ-2**





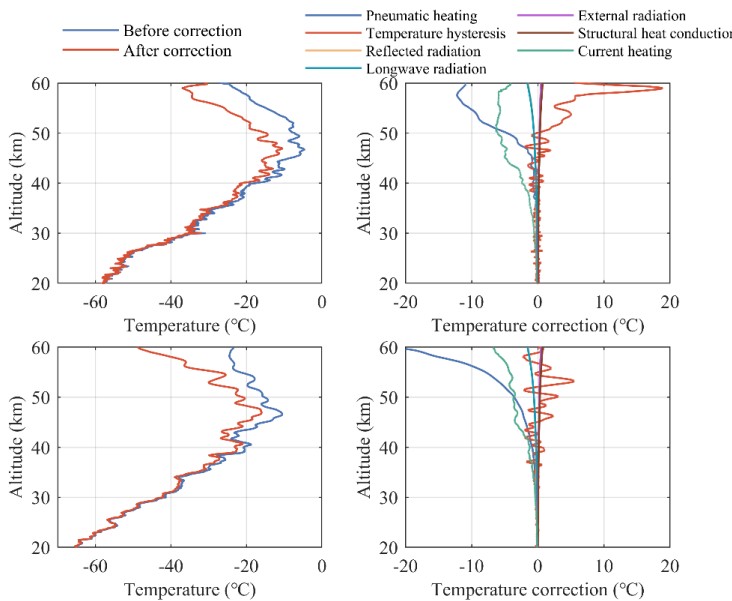


**Figure A2. The vertical distribution of (a) original and corrected temperature, and (b) each correction subterm.**

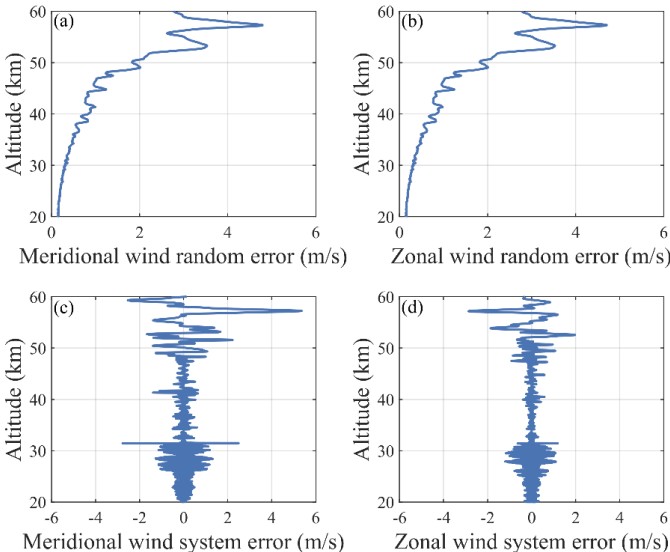


**Figure A3. Random error of (a) meridional wind and (b) zonal wind, and systematic error of (c) meridional wind**
**and (d) zonal wind for HJ-1**





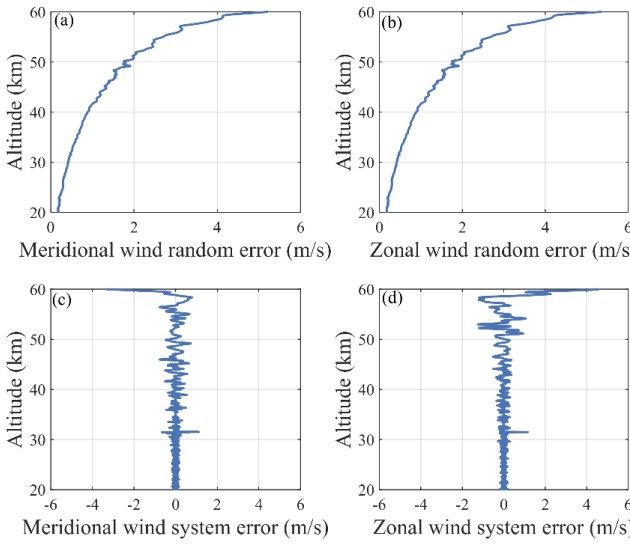


**Figure A4. Random error of (a) meridional wind and (b) zonal wind, and systematic error of (c) meridional wind and (d) zonal wind for HJ-2**

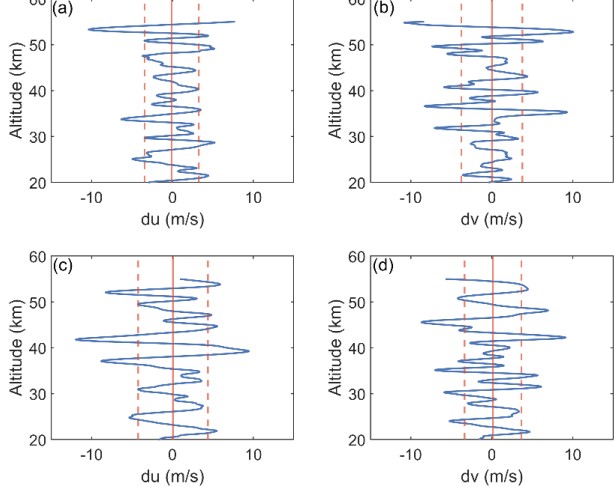


**Figure A5. (a) Zonal wind and (b) Meridional wind disturbance profile caused by GWs for HJ-1, and (c) Zonal wind and (d) Meridional wind disturbance profile caused by GWs for HJ-2. The solid and dashed lines represent the mean and standard deviation over the entire height, respectively**



**Code and data availability**

SABER data are available from ftp://saber.gats-inc.com/Version2_0/Level2A/ website, MERRA2 data are available from https://disc.gsfc.nasa.gov/ website. The data processing scripts and the rocket data are available from the first author upon reasonable request.

**Author contributions**

HM and SZ initiated the study. HY and HJP designed the scheme, HY analyzed data and drew figures, HY wrote the manuscript. All the authors interpreted results and revised the manuscript.

**Competing interests**

The contact author has declared that neither of the authors has any competing interests.

**Acknowledgments**

This work was supported by the National Natural Science Foundation of China (Grant no. 42405065) and the National Natural Science Foundation of China (Grant no. 42275060). Additionally, helpful comments by the editors and the specific anonymous reviewers are gratefully acknowledged.

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
