# Peer review of "Evaluation of middle atmosphere temperature and wind"

_EGUsphere, 2024_

## Referee Comment (RC2)

Review of "Evaluation of middle atmosphere temperature and wind measurements and and their disturbance characteristics by meteorological rockets" by He et al. submitted to *Atmospheric Measurement Techniques*

General comments:

This study obtained middle atmosphere temperature and wind measurements from 20-60 km in northwest China by two meteorological rockets. The detection results are compared with satellite, empirical model and reanalysis data, and the error analysis theory is carried out in combination with the of the drop sounding and atmospheric disturbance characteristics. The observations in the manuscript provide valuable results for understanding the atmospheric environment in the near space. Moreover, the authors, in combination with ERA5 data, have captured typical cases of gravity wave, which is a very interesting result. Although the authors' dataset and observation could be of high scientific value, the study in its present form suffers from several flaws and I recommend publication with suitable revisions.

Specific comments:

1. The theoretical background introduced in the introduction is not complete enough. Since a considerable part of the manuscript is about the analysis of atmospheric disturbance characteristics, the author needs to emphasize in the introduction the significance of atmospheric fluctuations in detection, or why the rocket detection results should be used to analyze the characteristics of gravity activities?

2. Effective height interval for analysis is from 20~60 km.
Does the rocket stop receiving data when it actually reaches a height of 20 kilometers (during the descent phase)? Why is only the data over 20 kilometers mentioned here? If the author analyzes the data from 0 to 20 kilometers together, more valuable results will be highlighted. Also, the maximum detection altitude of the rocket exceeds 70 kilometers (Figure 3).

3. The specific calculation process of atmospheric parameters shown in Figure 2 doesn't seem to correspond to the previous statement. For instance, shouldn't base pressure measurement be conducted independently? In the picture, it seems to be obtained after temperature correction. Similarly, it is recommended that the calculation formulas for air pressure and density be further clarified in the main text.

4. Errors and deviations in the result comparison.
Since authors evaluate the accuracy of the rocket detection results, a deviation analysis is conducted by comparing with the reference data, and at the same time, the error results are obtained through iterative calculation of the parameters during the detection process. The author should distinguish the differences between the two results in the text, and what are the respective functions of the given deviation results and error results?

5. Temperature correction.

Temperature correction is the key to ensuring the accuracy of temperature measurement. Judging from the results in the figure, the author's temperature correction results are reliable. However, the main text and supplementary materials lack further descriptions of the instrument's performance parameters, which makes it unclear to readers who focus on the measuring instrument itself the extent to which the instrument parameters affect the measurement. Similarly, the author provided the results of each temperature correction sub-term in the supplementary materials, which are very valuable measurement results. It is suggested that the author further explore and analyze the internal information.

6. Regarding the extraction of gravity waves, the author should provide the original profile of the rocket detection, the fitted background profile, and the disturbance profile, etc., to visually demonstrate whether the extracted waves are reasonable.

7. Regarding the slope of the gravity wave spectrum. The extraction of the slope of the gravity wave spectrum largely depends on the fitting interval. Due to the significant fluctuations of the power spectrum throughout the wavenumber range, different fitting intervals may cause obvious slope differences.

8. Based on the parameters obtained by the stokes parameter method, the author points out that the gravity waves of HJ-2 have a consistent propagation process. However, there is a significant change in the horizontal propagation direction (from 215° to -36°). The results in Figure 12 can demonstrate that within the 40-50km range, the propagation characteristics reflected by HJ-1 and HJ-2 through the stokes parameter method are consistent with the wave characteristics of ERA5. However, it cannot be determined whether the wave propagation characteristics at the entire altitude of the same rocket exploration are consistent (whether the wave in 40-50km and the wave in 20-50km are from the same source).

Minor comments:
L24: momentum deposited → momentum deposition

L91: The captions and the main text should be separated.

16. L59: The falling spheres can obtain the atmospheric density profile of 30-100 km, and then calculate the wind field, temperature and pressure, the thermistor measurement can obtain the atmospheric temperature from 20 to 60 km, and then calculate the density, pressure, and wind field → The falling spheres can obtain the atmospheric density profile of 30-100 km, and then calculate the wind field, temperature and pressure. The thermistor measurement can obtain the atmospheric temperature from 20 to 60 km, and then calculate the density, pressure, and wind field

L64: Atmospheric density are measured using GPS data on a rigid falling ball and found that the deviation from → Atmospheric density is measured using GPS data on a rigid

falling ball and the measured deviation from

L70: to obtain near space meteorological detection data from 20 to 60 km → to obtain meteorological detection data from 20 to 60 km

L91: One space is missing

L112: form → from

L389: are → is

Unfortunately, neither my abilities nor my time allow me to find all the grammatical problems throughout the manuscript. Therefore, I sincerely ask the authors to check the full text by themselves, and preferably seek advice from a native speaker.

---

## Author Comment (AC2)

**Response to Reviewer #2:**

General comments:

This study obtained middle atmosphere temperature and wind measurements from 20-60 km in northwest China by two meteorological rockets. The detection results are compared with satellite, empirical model and reanalysis data, and the error analysis theory is carried out in combination with the of the drop sounding and atmospheric disturbance characteristics. The observations in the manuscript provide valuable results for understanding the atmospheric environment in the near space. Moreover, the authors, in combination with ERA5 data, have captured typical cases of gravity wave, which is a very interesting result. Although the authors' dataset and observation could be of high scientific value, the study in its present form suffers from several flaws and I recommend publication with suitable revisions.

**Response**: Thank you for your understanding and recognition of our work. Without your help and advice, the manuscript would not have been significantly improved. We have carefully reviewed the comments and have revised the manuscript accordingly. Our responses are given in a point-by-point manner below. Changes to the manuscript are shown in the revised manuscript with "track changes".

Specific comments:

1. The theoretical background introduced in the introduction is not complete enough. Since a considerable part of the manuscript is about the analysis of atmospheric disturbance characteristics, the author needs to emphasize in the introduction the significance of atmospheric fluctuations in detection, or why the rocket detection results should be used to analyze the characteristics of gravity activities?

**Response**: Thank you for pointing out this issue. Based on your suggestions, we have made the following modifications:

L40 Added "Atmospheric disturbances, as the superposition of waves at different scales (including turbulence, gravity waves, planetary waves, etc.), are one of the main dynamic processes in the near space. As the height increases, the density decreases exponentially. The amplitude of these disturbances, such as gravity waves, gradually increases during the upward propagating process, and the impact of the wave becomes more and more significant (Lindzen, 1981; Alexander et al., 2010)"

L61 Added "Due to the large amplitude of atmospheric waves in this height range, the momentum and energy dissipated by wave fragmentation can cause drastic changes in meteorological elements such as wind field, density and temperature in the surrounding atmosphere. Therefore, analyzing the interaction mechanism between atmospheric wave and background atmosphere has always been one of the important research directions of in-situ observational data."

2. Effective height interval for analysis is from 20~60 km.

Does the rocket stop receiving data when it actually reaches a height of 20 kilometers (during the descent phase)? Why is only the data over 20 kilometers mentioned here? If the author analyzes the data from 0 to 20 kilometers together, more valuable results will be highlighted. Also, the maximum detection altitude of the rocket exceeds 70 kilometers

(Figure 3).

**Response**: In fact, the point you put forward is completely correct. During the actual rocket launch, the highest point of the trajectory was at an altitude of over 70 kilometers. However, in the early stage of the separation of the rocket and the parachute, if the descent speed is too fast, the sensor cannot detect effective data (the temperature correction processing in the later stage of the high-altitude part also has a prerequisite, that is, the descent speed of the radiosonde is lower than a certain threshold). Therefore, in the rocket we use, the selection of 60 kilometers as the upper limit of the effective detection altitude is determined based on the performance of the instrument itself, taking into account the range of the falling speed. Similarly, as can be seen from Figure 1, the data return transmission requires a ground receiver. As can be seen from Figure 3, as the falling height decreases, the horizontal drift distance becomes increasingly farther. After the altitude dropped to 20 kilometers, the receive stop receiving (taking into account both time and data quality). The data analysis range in this manuscript is 20 to 60 kilometers, which is also the standard detection height range for this model of product.

3. The specific calculation process of atmospheric parameters shown in Figure 2 doesn't seem to correspond to the previous statement. For instance, shouldn't base pressure measurement be conducted independently? In the picture, it seems to be obtained after temperature correction. Similarly, it is recommended that the calculation formulas for air pressure and density be further clarified in the main text.

**Response**: Thank you for pointing out this issue. Based on your suggestions, we have made the following modifications:

Figure 2 was redrawn:

[Figure]

**Figure 2. Atmospheric parameters calculation process.**

L106 Added "The air pressure at each height layer is calculated from the measured base point air pressure (20 km) using the pressure height formula:

$$P = P_{\text{d}} \exp \frac{-g_0(H - H_{\text{d}})}{R^* T_{\text{d}}}$$

Among them, $P$ represents the air pressure at the calculated height, $P_{\text{d}}$ is the air pressure of the adjacent lower layer, $H$ is the geopotential, $H_{\text{d}}$ is the geopotential of the adjacent lower layer, $R$ is the dry air gas constant, and $T_{\text{d}}$ is the temperature of the adjacent lower layer. Given the temperature and air pressure, the atmospheric density can be calculated through the ideal gas state equation."

4. Errors and deviations in the result comparison.

Since authors evaluate the accuracy of the rocket detection results, a deviation analysis is conducted by comparing with the reference data, and at the same time, the error results are obtained through iterative calculation of the parameters during the detection process. The author should distinguish the differences between the two results in the text, and what are the respective functions of the given deviation results and error results?

**Response**: In the manuscript, the deviation result describes the gap between the rocket detection result and the reference value (satellite, reanalysis, etc.). The greater the deviation, the more it deviates from the reference data. The purpose is to demonstrate the degree of consistency (deviation) between the rocket detection results and other data, and it is a comparison between different data. The error result describes the measurement error of the rocket measurement instrument itself (including systematic error and random error), which is a comparison among different heights and sub-items within the rocket detection results. Based on your suggestions, we have made the following modifications:

L215: Added "Here, we use the deviation result describes the gap between the rocket detection data and the reference value (satellite, reanalysis, etc.). The purpose is to demonstrate the degree of consistency or deviation between the rocket detection results and other data, and it is a comparison between different data."

L249 Added "Accurate measurement is the prerequisite for conducting further data analysis and application. To further analyze the sources of deviation between the rocket detection results and other data, as well as the reliability of the disturbance analysis, the error level of the rocket instrument is discussed here, which is a comparison among different heights and sub-items within the rocket own detection results."

5. Temperature correction.

Temperature correction is the key to ensuring the accuracy of temperature measurement. Judging from the results in the figure, the author's temperature correction results are reliable. However, the main text and supplementary materials lack further descriptions of the instrument's performance parameters, which makes it unclear to readers who focus on the measuring instrument itself the extent to which the instrument parameters affect the measurement. Similarly, the author provided the results of each temperature correction sub-term in the supplementary materials, which are very valuable measurement results. It is suggested that the author further explore and analyze the internal information?

**Response**: Based on your feedback, we have provided supplementary explanations regarding the information of the instrument itself. We have supplemented the detailed description of the sensor and its performance parameters in the manuscript. We have drawn

the internal structure of the radiosonde so that readers can have a more intuitive understanding of the instrument information. At the same time, the information of the telemetry requirements is supplemented.

L79: The rocket radiosonde is mainly composed of temperature sensors, pressure sensors, satellite navigation and positioning modules, data acquisition circuits, transmitters, wireless remote control modules, batteries, switches, fixed frames, insulation boxes and fiberglass reinforced plastic shells, etc. The temperature sensor adopts a bead thermistor, purchased from the shelf, model MF51MP-D (Blue Crystal Electronics). The pressure sensor adopts a high-precision digital pressure sensor, purchased from the shelf, model ms5607 (Switzerland). The navigation and positioning module adopts the high-precision positioning module of Beidou, and the antenna uses a four-arm helical antenna, which is a customized product. The main MCU of the data acquisition circuit adopts a 32-bit processor with ARM core, featuring low power consumption and mixed signal processing capabilities. It has a 14-bit A/D conversion accuracy, which can meet the measurement accuracy requirements of sensors. The digital transmitter is composed of dedicated RF chips and power amplifier modules to form a frequency point digital transmitter. It has the advantages of small size and adjustable frequency. When used in conjunction with ground receiving equipment, it can achieve data transmission within a diagonal distance range of 200 kilometers. The physical appearance and structural layout of the rocket sounding instrument are shown in Figure 1, and the main performance indicators of the rocket sounding instrument are shown in the table A1.

[Figure]

**Figure 1. (a) The physical appearance and (b) structural layout of the rocket sounding instrument.**

Table A1. Main performance indicators of rocket radiosondes

| Indicator name | Performance parameters |
|---|---|
| Transmitter frequency | 400MHz ~ 406MHz |

| | | |
|---|---|---|
| Carrier frequency stability | ±20kHz | |
| Emission spectral width | ≤50kHz(-50dB) | |
| Transmitter power | 100mW ~ 200mW | |
| Digital signal transmission mode | GFSK | |
| Data transmission rate | 4800bps | |
| Data update rate | ≥2Hz | |
| Positioning accuracy | Horizontal direction | 5m (CEP 90%) |
| | Vertical direction | 5m (CEP 90%) |
| | Speed | 0.2m/s (CEP 90%) |
| Temperature | Measurement range | -90°C ~ +55°C |
| | Static calibration accuracy of the sensor | ≤±0.2°C |
| | Resolution | 0.1°C |
| pressure | Measurement range | 1060hPa ~ 5hPa |
| | Static calibration accuracy of the sensor | ≤±0.8hPa |
| | Resolution | 0.1hPa |

As for each temperature correction sub-term in the supplementary materials, we have made the following supplementary discussions:

L233 Added "Among the various correction sub-items for rocket detection temperature, the influence degree of pneumatic heating, current heating, and temperature hysteresis are relatively large, and these influences gradually decrease as the height decreases overall."

6. Regarding the extraction of gravity waves, the author should provide the original profile of the rocket detection, the fitted background profile, and the disturbance profile, etc., to visually demonstrate whether the extracted waves are reasonable.

**Response**: In this paper, we need to explore the wave characteristics in different height intervals. If hodograph method is used for analysis, many intervals do not meet the characteristics of monochromatic waves (Figure 10 already shows that gravity waves are polychromatic waves). Therefore, we use Stokes parameters to extract the parameters of gravity waves.

L311: Added "Taking HJ-2 as an example, the extraction process of zonal wind and meridional wind disturbances is demonstrated (Figure A6)."

[Figure]

**Figure A6. The disturbance extraction process of (up) zonal wind and (down) meridional wind for HJ-2**

From the figure, an intuitive phenomenon can be observed: before high-pass filtering, there is a vertical wavelength of 20km (b), and after high-pass filtering, the wavelengths are all within 10km. This also indicates the correctness of data processing.

7. Regarding the slope of the gravity wave spectrum. The extraction of the slope of the gravity wave spectrum largely depends on the fitting interval. Due to the significant fluctuations of the power spectrum throughout the wavenumber range, different fitting intervals may cause obvious slope differences.

**Response**: Authors strongly agrees with your view that extraction of the slope of the gravity wave spectrum largely depends on the fitting interval, since different fitting intervals may cause obvious slope differences. At the same time, considering the existence of noise, the slopes in spectra demonstrated in the manuscript cannot be interpreted so directly.

So, based on your suggestion, we have removed the description of the FFT spectral results here because their interpretation (especially the slope) may not be entirely accurate, and the Lomb-Scargle spectral results have already achieved the author's discussion purpose: to analyze the dominant vertical wavelengths in the two detections.

8. Based on the parameters obtained by the stokes parameter method, the author points out that the gravity waves of HJ-2 have a consistent propagation process. However, there is a significant change in the horizontal propagation direction (from 215° to -36°). The results in Figure 12 can demonstrate that within the 40-50km range, the propagation characteristics reflected by HJ-1 and HJ-2 through the stokes parameter method are consistent with the

wave characteristics of ERA5. However, it cannot be determined whether the wave propagation characteristics at the entire altitude of the same rocket exploration are consistent (whether the wave in 40-50km and the wave in 20-50km are from the same source).

Response: Thank you for pointing out this issue. The author had mentioned in the manuscript the inconsistency between the local and overall wave propagation directions, and believed that this might be related to significant wind speed changes near 40 km. However, the interaction mechanism between gravitational waves and the background field is complex. Changes in the wind field may lead to alterations in the propagation direction, but at the same time, if new wave sources are generated, it will also cause inconsistent propagation directions at different heights. Based on the current data of this manuscript alone, it is impossible to verify whether they come from the same wave source (this does not affect the key point of perturbation analysis: the judgment of wave fragmentation). Therefore, to ensure the rigor of the expression, the author has made the following modifications:

L396 : Deleted "The wave propagation direction of HJ-2 is significantly different in the entire and local ranges, possibly due to significant wind speed changes near 40 km (Figure 5c)."

Minor comments:

1. L24: momentum deposited → momentum deposition

Response: Thank you for pointing out this issue. Based on your suggestions, we have made the corresponding modifications.

2. L91: The captions and the main text should be separated

Response: Thank you for pointing out this issue. Based on your suggestions, we have made the corresponding modifications.

3. L59: The falling spheres can obtain the atmospheric density profile of 30-100 km, and then calculate the wind field, temperature and pressure, the thermistor measurement can obtain the atmospheric temperature from 20 to 60 km, and then calculate the density, pressure, and wind field → The falling spheres can obtain the atmospheric density profile of 30-100 km, and then calculate the wind field, temperature and pressure. The thermistor measurement can obtain the atmospheric temperature from 20 to 60 km, and then calculate the density, pressure, and wind field

Response: Thank you for pointing out this issue. Based on your suggestions, we have made the corresponding modifications.

4. L64: Atmospheric density are measured using GPS data on a rigid falling ball and found that the deviation from → Atmospheric density is measured using GPS data on a rigid falling ball and the measured deviation from

Response: Thank you for pointing out this issue. Based on your suggestions, we have made the corresponding modifications.

5. L70: to obtain near space meteorological detection data from 20 to 60 km → to obtain meteorological detection data from 20 to 60 km

**Response**: Thank you for pointing out this issue. Based on your suggestions, we have made the corresponding modifications.

6. L91: One space is missing

**Response**: Thank you for pointing out this issue. Based on your suggestions, we have made the corresponding modifications.

7. L112: form → from

**Response**: Thank you for pointing out this issue. Based on your suggestions, we have made the corresponding modifications.

8. L389: are → is

**Response**: Thank you for pointing out this issue. Based on your suggestions, we have made the corresponding modifications.